# Can the current radiation dose, in chest tomosynthesis, be reduced with retained image quality? A study in the context of lung cancer screening programs

Masoud Jadidi[1]*, Angelica Svalkvist[2,3], Magnus Båth[2,3], Sven Nyrén[1,4]

**1** Department of Molecular Medicine and Surgery, Karolinska Institutet, Stockholm, Sweden,
**2** Department of Medical Physics and Biomedical Engineering, Sahlgrenska University Hospital, Gothenburg, Sweden, **3** Department of Medical Radiation Sciences, Institute of Clinical Sciences, University of Gothenburg, Gothenburg, Sweden, **4** Department of Radiology, Karolinska University Hospital, Stockholm, Sweden

* masoud.jadidi@ki.se

## Abstract

### Objective

Digital chest tomosynthesis (DTS) is a low-cost and low- dose imaging technique that has not yet reached general use. It might, however, be well suited to lung cancer screening programs. In lung cancer screening, it is essential to keep radiation dose low, as a large number of healthy individuals have to be examined to detect one case of lung cancer. Lung cancer screening has been proven to lower deaths in lung cancer by approximately 20%, but this limited success comes at high cost. We believe that adding interval examination with DTS to lung cancer screening programs could increase survival and reduce cost, without significantly increasing the radiation dose to the individual. We therefore conducted this study to investigate if the current radiation dose, in chest tomosynthesis, could be reduced with retained image quality.

### Methods

50 patients with known or suspected lung malignancy were imaged with chest X-ray equipment using the tomosynthesis option. Three DTS examination protocols were applied for each patient. One DTS examination was performed with the dose level as in the vendor-recommended protocol, and two other examinations were performed with low-dose protocols, reducing the exposure dose by 30% and 50%, respectively. Images from all three protocols were processed using the vendor-recommended post-processing settings, including a filtered back projection algorithm. The images generated by the DTS protocols were evaluated by four observers anonymously and in random order. The observers rated the quality of the reconstructed section images according to predefined quality criteria for different thoracic regions. Visual grading

**Data availability statement:** The dataset containing imaging protocol settings, de-identified patient data, corresponding dose calculations, ViewDEX configuration files, and the observers' evaluation results have been made publicly available in the Figshare repository, where it has been assigned the DOI: https://doi.org/10.6084/m9.figshare.30621287.

**Funding:** The author(s) received no specific funding for this work.

**Competing interests:** No competing intrerests.

characteristics (VGC) were used to analyze the data, and the area under the VGC curve ($AUC_{VGC}$) was used as a figure-of-merit.

## Results

The result of VGC analysis showed no reduction in imaging quality for the protocol with a 30% dose reduction compared with the vendor-recommended protocol for any of the classes of criteria, namely demarcation ($AUC_{VGC} = 0.51$, $p = 0.57$), disturbance ($AUC_{VGC} = 0.51$, $p = 0.59$) bone structure ($AUC_{VGC} = 0.50$, $p = 0.88$), and tumor homogeneity ($AUC_{VGC} = 0.47$, $p = 0.09$). For the protocol with a 50% dose reduction, image quality was reduced for the classes of criteria demarcation ($AUC_{VGC} = 0.47$, $p = 0.03$) and disturbance ($AUC_{VGC} = 0.47$, $p = 0.02$) compared with the vendor-recommended protocol, while no quality reduction was found for the classes of bone structure ($AUC_{VGC} = 0.50$, $p = 0.79$) and tumor homogeneity ($AUC_{VGC} = 0.46$, $p = 0.10$).

## Conclusions

This study has not shown any reduction in image quality from reducing the radiation dose by 30%, compared with the vendor-recommended protocol, which might indicate that such a reduction could be used in clinical practice.

## Introduction

Digital chest tomosynthesis (DTS) is a technology that has been employed alongside the introduction of flat panel detectors [1]. The technology has never achieved widespread clinical use, even though it offers obvious advantages, such as improved diagnostic accuracy compared to standard chest X-rays.

The main advantages of DTS over computed tomography (CT) are lower cost, shorter examination time, and lower radiation dose [2–4], and these advantages might be of great importance in a lung screening program. Lung lesions can be detected and localized with DTS almost as well as with low-dose computed tomography (LDCT) [5–7].

Lung cancer is the leading cause of cancer mortality in the industrialized world, accounting for an estimated 18% of total cancer deaths worldwide. Furthermore, lung cancer is one of the most frequently occurring cancers and is the leading cause of cancer death in men, and the second most commonly diagnosed cancer and leading cause of cancer death in women, after breast cancer [8,9]. Studies have shown that smoking is the main risk factor for lung cancer, and smokers increase the risk of developing lung cancer by 30 times when compared to nonsmokers [10–12].

There is very strong evidence to show that detecting lung cancer at an earlier stage reduces cancer-related mortality. This has led to the conclusion that efforts should be made to detect early-stage lung cancer in non-symptomatic individuals [9,13–15]. Studies on early detection have been conducted since the 1970s using different methods such as conventional chest X-ray, sputum cytology, and incidentally

detected lung nodules by CT and LDCT. Early screening studies using chest X-ray (CXR) found limited benefit regarding reduction in mortality, so CT has been the preferred method for lung cancer screening since the 1990s. Use of LDCT dates back almost three decades [9,13–18].

A number of programs using CT for lung cancer screening have been initiated but are hampered by high costs. The cost of saving one (life quality-adjusted), year of life in 2021 was estimated to be 72,564 US dollars (ranging between 59,493 – and 85,837 dollars), making cost effectiveness an important issue [15].

During lung cancer screening programs, a large number of individuals who do not have cancer have to be exposed to radiation to detect one positive case. This leads to a large number of follow-up examinations, making screening programs significant contributors to population doses of radiation, even though the dose from each single examination is low. Another effect of the high cost of CT examination is that intervals between examinations are kept at two years. This does not, however, prevent patients from developing fatal lung cancer in the interval [19,20]. Two large randomized controlled trials, the National Lung Screening Trial (NLST) with 53,454 participants in the USA (between 2002 and 2004), and the Netherlands Leuvens Screening Onderzoek (NELSON) with 15,792 participants in the Netherlands (between 2003 and 2006), showed a reduced lung cancer mortality of 20% in a high-risk population after screening with LDCT in the NLST trial, and a cumulative reduction of 24% at 10 years in the screening group for men. This means that a large proportion of the screening population will still die from lung cancer [9,19,21,22].

We have only found one study where DTS has been used in lung cancer screening of 2000 patients. This study showed a detection rate of 0.9% lung cancer in the screening population, which is comparable to the rate detected in CT lung cancer screening programs [23].

There are also some challenges, such as socio-economic cost, radiation burden on the population, resource requirements in terms, for example, of radiologists to meet the needs of screening programs, and the risk of overdiagnosis. Therefore, few countries have been able to organize and implement large lung cancer screening programs, but many countries have launched pilot projects or screening programs on a smaller scale [9,22,24].

Ferrari et al. investigated the radiation dose from three different types of chest examinations, conventional radiography, tomosynthesis, and CT. The dose in posterior-anterior projection in conventional chest radiography was 0.01 mSv, and in lateral projection was 0.15mSv. The posterior-anterior dose in chest tomosynthesis was 0.1–0.2 mSv, for conventional CT 4–8 mSv, and for the low-dose CT was 1.5 mSv [25]. A previous study has reported that the effective dose for DTS, for a standard-sized patient (170 cm-70 kg), is ~ 0.13 mSv, including scout view, which is only ~2% of that for an average chest CT and only two to three times the effective dose from conventional chest radiography examination [26]. Several studies have introduced different methods for further reduction of radiation dose for 3D medical images, including tomosynthesis, such as sparse sampling, varied tube voltage reduction and dose ratio, shutter scan acquisition and model-based iterative reconstruction (MBIR) which is the most efficient method for dose reduction [27–32].

Furthermore, CT might not necessarily be a cost-effective imaging choice, when compared to chest tomosynthesis [33]. From an economic perspective, studies have shown that the cost of a chest tomosynthesis examination is approximately 35% and in some cases 17% of that for a CT examination, meaning a significant cost saving for diagnostic imaging [34,35]. From a clinical perspective, scientific studies have shown that DTS can reduce utilization of CT in 70%−80% of clinical cases with suspected lesions on CXR [36–38]. Additionally, DTS has detection rates of 49%−58% and 48%−62% relating to solid pulmonary nodules 5 mm or larger and 6 mm or larger respectively, and meets the criteria for follow-up after CT examination [39].

Söderman et al. [40] investigated the impact of dose level on precision of measurements on simulated nodules in DTS and declared that DTS can be used to detect the growth of pulmonary nodules. The study showed that a dose reduction from the standard level did not significantly affect the accuracy and precision of nodule size measurements. However, the ability to detect nodule growth decreased with smaller nodule sizes and lower dose levels.

A previous study by Asplund et al. [41] showed no significant differentiation in detectability of pulmonary nodules by adding artificial noise to DTS images from eighty-six patients, simulating a dose level corresponding to 32% of the original dose.

To the best of the authors' knowledge, no published patient studies on real patients have investigated the impact of reducing the original dose level for chest tomosynthesis examinations, in a clinical setting.

The aim of this study was to assess the image quality of low-dose tomosynthesis compared to the vendor-recommended protocol with the standard dose. It is a prospective observational study of image quality with four observers using visual grading characteristics. The purpose is to assess the effect of radiation dose reduction and evaluate the possibility of lowering the dose where DTS is part of a lung cancer screening program.

## Methods and materials

### Patient data

The data used in this study contain potentially identifying and sensitive patient information, including patient names, patient identification numbers, and other confidential clinical details. For this reason, full access to the dataset is restricted. However, the dataset containing imaging protocol settings, de-identified patient data, corresponding dose calculations, ViewDEX configuration files, and the observers' evaluation results have been made publicly available in the Figshare repository, where it has been assigned the DOI: httpsdoi.org10.6084m9.figshare.30621287.

Approval for the study was obtained from the Regional Ethical Review Board in Stockholm, Sweden. The data are owned and controlled by Karolinska University Hospital, Solna, in collaboration with Karolinska Institutet, Stockholm. These institutions impose the restrictions on data sharing to ensure the protection of patient privacy and compliance with Swedish and EU ethical and data-protection regulations. In addition, informed consent, (both verbally and signed and dated consent forms) was obtained from all participants in the study. From March 31, 2021 to June 1, 2023, 50 patients (24 men and 26 women) with an average age of 66 years (range 24–81 years), average weight of 71 kg (range 45–119 kg) and average height of 158 cm (range 155–191 cm) with known pulmonary malignancies were included in the study. During the same period CT examinations from the patients were retrospectively collected and were used as reference for the tomosynthesis examinations.

Inclusion criteria were patients above 24 years of age, with at least one well visualized lung lesion on CT-examination.

An even gender distribution and variation in body size were sought. Patients with mental illness or dementia were excluded when they were judged to be unable to make an informed choice. The selected patients were scheduled for DTS examination by the author during a weekend when the tomosynthesis examination could easily be carried out without disturbing daily operation at the Department of Radiology. To avoid changes in lesions volume between CT and DTS examinations, the maximum time between the CT and DTS examination was two weeks, without any treatment or surgical intervention during the interval.

Information about the study and the examination procedure was explained by phone, and after acceptance by the participant, the DTS examination was scheduled based on the participant's availability.

For study purposes, three DTS series were performed with different exposure settings, acquired on each patient.

### The DTS system and data acquisition

With the DTS technique, individual and short-duration exposures take place as the X-ray tube moves within a limited angular range from one end of the X-ray flat panel detector to the other, and the detector acquires an individual image for each exposure. The collected images resulting from the DTS acquisition can be reconstructed into multiple coronal section images with different depths of examination volume using filtered back projection. The reconstructed section images contain much less disturbing overlying anatomy than conventional projection images [42–47].

The chest tomosynthesis examinations were performed on the recently upgraded GE Healthcare DTS system, Definium 8000 system with VolumeRAD data acquisition technology, acquired using an indirect conversion flat-panel

detector with a source-to-image distance (SID) of 180 cm and with automatic exposure control (AEC) for the scout projection and tomosynthesis acquisition.

The vendor recommended acquisition protocol includes a tube voltage of 120 kV and a total filtration of 3 mm Al + 0.1 mm Cu. According to the GE vendor-recommended protocol for chest tomosynthesis examination, the X-ray tube performs a continuous vertical motion, acquiring 60 projections images in the angular interval of ±15° during a time period of approximately ten seconds. During the vertical motion of the X-ray tube, the detector position was fixed.

The material for this study consists of three tomosynthesis protocols per patient. In order to achieve dose reduction of 30% and 50%, respectively, and to be able to compare the average energy in the X-ray spectrum at the detector between three DTS protocols, it was necessary to adjust the X-ray conditions.

The exposure parameters for tomosynthesis acquisition requires an initial scout image (a conventional PA projection) acquired using automatic exposure control (AEC). The tube load (mAs) value obtained from the scout image is then multiplied by a dose ratio (commonly 10) and the resulting total mAs is evenly divided between the 60 projection images. Due to a technical limitation of the X-ray system limiting the exposure to minimum 0.25 mAs per projection [40,48], changing the dose ratio would not lead to lower patient exposure. To overcome this limitation, an additional 20 mm aluminum (Al) was placed in front of the collimators during collection of the images. In this way the output from the X-ray tube was increased to a level where a change of dose ratio resulted in reduced output from the X-ray tube. The addition of extra Al filtration will change the X-ray spectrum and lead to an increase in average photon energy. A spectrum processor was used to simulate the X-ray spectra (Siemens Healthiness, Spectrum Simulation Tool). The mean photon energy of the original spectrum (120 kV, 13 mm Al, 0.1 mm Cu) was 67.4 keV and for the new protocol (100 kV, 33 mm Al, 0.1 mm Cu), the mean photon energy was 67.5 keV.

To minimize this effect the tube voltage was reduced to 100 kV (instead of 120 kV) when the extra Al filtration was added. An additional 20 mm of aluminum filtration was implemented identically for all three tomosynthesis protocols, resulting in comparable radiation quality across these settings. The resulting beam quality, including the mean photon energy, was therefore effectively the same for all tomosynthesis acquisitions, so that the differences observed between protocols are not attributable to changes in spectral quality but to differences in dose and acquisition geometry.

After addition of the extra Al filtration the dose ratio was varied to obtain different levels of dose reduction, where dose ratio 10 was used for the full dose protocol (chest tomosynthesis 1), dose ratio 7 for the protocol aiming at 30% dose reduction (chest tomosynthesis 2) and dose ratio 5 för the protocol aiming at 50% dose reduction (chest tomosynthesis 3). Other settings such as collimation of the X-ray tube, source-to-detector distance (SDD), X-ray tube swing angle, and total exposure time, the number of exposures, image post-processing parameters, and reconstruction conditions were the same for all three protocols. Reconstruction algorithm filtered back projection (FBP) was used in order to reduce the blurred out-of-plane anatomy [42], which was recommended by the vendor. To facilitate estimation of patient radiation dose in chest tomosynthesis, a conversion coefficient of (0.26 mSv/(Gy·cm²) was introduced that links the total DAP of the tomosynthesis examination to the corresponding effective dose. This conversion coefficient was obtained for a standard-sized adult patient using Monte Carlo simulations of the VolumeRAD system (GE Healthcare, Chalfont St. Giles, UK), which at the time was the most extensively evaluated platform for both scientific and clinical studies of chest tomosynthesis.

Effective dose for DTS was calculated by multiplying the DAP of the DTS examination by a conversion factor of 0.26 mSv Gy⁻¹ cm⁻² (25). Because the DAP of the DTS examination is not stored with the images, it was derived using a validated method developed by Båth et al. [48], in which the DAP is estimated from data contained in the digital imaging and communications in medicine (DICOM) header of the scout image [26,48].

In this cohort, 52% of participants had BMI < 25, 36% had BMI 25–30, and 12% had BMI > 30, indicating a broad distribution of body habitus. Table 1 shows that both DAP and effective dose increase across the BMI groups. Effective doses are lower when using ratios 7 and 5 but follow the same BMI-dependent pattern, confirming that higher BMI is generally

Table 1. Summary of dose-area product (DAP), effective dose, and percentage of effective dose in relation to ratio 10, estimated across body mass index (BMI) groups using different ratios.

| BMI | Average DAP valuedGy/cm2 | SD- DAP value | Ratio 10 | | Ratio 7 | | | Ratio 5 | | |
| | | | Average effective Dose (Ratio 10)mSv | SD effective dose (Ratio 10) | Average Effective Dose (Ratio 7)mSv | Percentage of effective dose in relation to ratio 10 | SD effective dose (Ratio 7) | Average Effective Dose (Ratio 5)mSv | Percentage of effective dose in relation to ratio 10 | SD effective dose (Ratio 5) |
|---|---|---|---|---|---|---|---|---|---|---|
| <25 | 2.53 | 0.73 | 0.56 | 0.03 | 0.39 | 70% | 0.13 | 0.29 | 52% | 0.10 |
| 25-30 | 4.11 | 1.03 | 0.9 | 0.04 | 0.60 | 67% | 0.14 | 0.43 | 48% | 0.11 |
| >30 | 4.78 | 0.57 | 1.01 | 0.03 | 0.63 | 62% | 0.22 | 0.45 | 45% | 0.18 |

associated with greater radiation exposure, consistent with the need for increased dose in patients with greater body thickness. This trend is confirmed by the percentage of effective dose in relation to ratio 10, with slight variation across the BMI groups. The effective dose conversion 0.26 mSv/Gycm2 was used for all ratio factors and all BMI groups.

## Examination workflow

For each patient, three acquisitions in the posterior-anterior (PA) direction in an upright position were performed in the order that was programmed in the DTS workstation and then reconstructed into coronal section image series. The patient was instructed to hold their breath during the acquisitions. According to the author who performed all the examinations, none of the patients had difficulty holding their breath during acquisitions. The reason for this was that the radiographer needed a couple of minutes to prepare for the next acquisition by restoring the position of the X-ray tube and changing the ratio value. During that time, the patient had time to recover their breath and relax.

The reconstructed section images at intervals of 5 mm in about 53 coronal images, were transferred automatically to the reading workstation, picture archiving and communication systems (PACS) (Sectra Medical Systems, Linköping, Sweden) for evaluation.

The estimated effective dose for the three protocols was as follows: for the vendor-recommended protocol 0.13 mSv (range 0.04–0.25 mSv), for the protocol with a 30% reduction 0.09 mSv (range 0.03–0.15 mSv), and for the protocol with a 50% dose reduction 0.06 mSv (range 0.02–0.12 mSv) (Fig 1).

## Observers' assessment of image quality

In total there were three DTS series for each patient, resulting in a total of 150 image series for the 50 patients.

In the present visual grading study, the software ViewDEX version 3.2 was used for evaluation of the quality of reconstructed DTS series. ViewDEX is an image viewer compatible with Digital Imaging and Communication in Medicine (DICOM) to facilitate image assessment and observer performance studies. The images were presented on a PACS monitor with RadiForce®, RX340, 3MP, Color LCD from Eizo (Hakusan, Japan), which is calibrated on a yearly basis.

Each observer evaluated image series when the image series were shown in a unique random order based on a set of criteria, with the option of giving a rating for each criterion. The answers of each observer were stored in a separate log file for analysis. In this study, the evaluation was based on absolute visual grading. This means that a subjective assessment was made by the observer using a rating scale between 1–6 where the image series were assessed one at a time on an absolute scale [49–54].

The observers (two senior consultant radiologists, one a specialist in radiology and one last- year resident) had access to a training set, including seven tomosynthesis test cases, before they started evaluation of the image series. They were able to go through the training set as many times as they wanted. The image evaluation questions in this study were

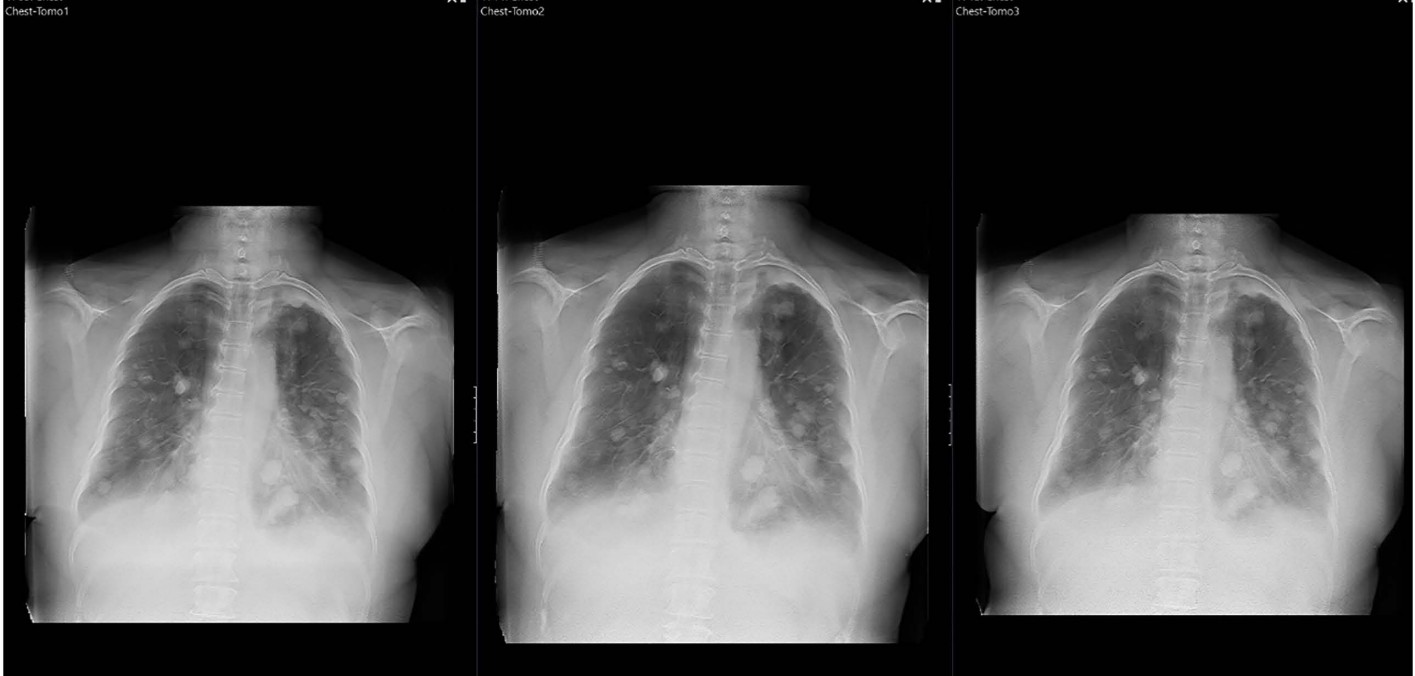

**Fig 1. Tomosynthesis series acquired with varying radiation doses: Chest Tomo 1 (standard dose), Chest Tomo 2 (30% dose reduction), and Chest Tomo 3 (50% dose reduction).**

designed based on the European guidelines on quality criteria for diagnostic radiographic images and CT [55–56], and the quality criteria developed in previous studies [1,27,44,57]. The criteria applied to image quality in pre-determined anatomical regions, were much the same as in previous publications [44,57]. The image quality ratings were based on a total of 14 questions, which included 5 about vessel demarcation, 5 concerning vessel disturbance, 2 related to bone structure, and 2 focused on tumor homogeneity (Table 2). No ratings were missing. All observers completed the full set of questions for each case. In ViewDEX, the image series are displayed in a unique random order for each observer along with a set of criteria, and the responses of the observer (the given rating for each criterion) are registered and stored in a log file for analysis. The randomization of image presentation order is managed automatically by ViewDex software, ensuring that the sequence assignment is free from human bias. Prior to the main evaluation, all observers received standardized instructions and reviewed a training set of representative test cases, which ensured familiarity with the rating criteria and evaluation procedure. This approach aligns with best practices in observer performance studies to ensure unbiased and reproducible results.

The main aim of using these criteria is to be able to evaluate the entire thoracic area. The image quality was evaluated in five different atomical regions, 1) basal-dorsal, 2) retro cardiac, 3) basal-ventral, 4) peripheral vessels and 5) the bone structure in spine and ribs. In addition, tumor homogeneity was assessed.

All observers evaluated the images, blindly, from each series and rated the image quality according to each criterion on a five-step ordinal scale: 1) Very bad, 2) Bad, 3) Fair, 4) Good, and 5) Very good.

## Statistical analysis

For the statistical analysis, the observers' ratings for the different criteria for the main classes of criteria, – demarcation, disturbance, bone structure and tumor homogeneity – were combined. For the purpose of analyzing the data, VGC (visual

**Table 2. The criteria to rate the image quality of the reconstructed digital tomosynthesis (DTS) image series.**

| Criteria | Structure |
|---|---|
| Vessel demarcation | Q1. Basal dorsal right side (small vessels)<br>Q2. Retro cardiac (small vessels)<br>Q3. Basal ventral right side (small vessels)<br>Q4. Basal ventral left side (small vessels)<br>Q5. Peripheral vessels, (upper, lateral right, small vessels) |
| Disturbance of vessels | Q6. Basal dorsal right side (disturbance of vessels in front/behind)<br>Q7. Retro cardiac, (disturbance of vessels in front/behind)<br>Q8. Peripheral vessels, (upper, lateral right, disturbance of vessels in front/behind)<br>Q9. Basal ventral right side, (disturbance of vessels in front/behind)<br>Q10. Basal ventral left side, (disturbance of vessels in front/behind) |
| Bone structure | Q11. Definition of bone structures in the spine<br>Q12. Definition of bone structure in the ribs |
| Tumor homogeneity | Q13. Internal tumor homogeneity<br>Q14. Tumor border definition |

grading characteristics) analysis was used as a statistical method, where two protocols are compared by creating a VGC curve. VGC analysis is a non-parametric, rank-invariant statistical method where the ratings for two protocols at a time (the test protocol and the reference protocol) are compared by creating a VGC curve. This is similar to how the ratings for normal and abnormal cases in receiver operating characteristics (ROC) analysis are used to generate an ROC curve. The analysis allows for a multi-reader, multi-case (MRMC) setup, where multiple observers and multiple cases are included and all observers assess all cases.

Correlated data are thus appropriately handled, similarly to MRMC ROC analysis. The resulting VGC curve represents the average ratings, and the asymmetric 95% confidence interval for AUCVGC is estimated via resampling. The p-value is calculated as the fraction of bootstrap samples producing results equal to or more extreme than the null hypothesis (AUCVGC = 0.5) [58].

Just as in MRMC ROC analysis, no comparison between the direct ratings by the observers is made in VGC analysis. Instead, the ratings from a given observer are used to determine an AUC for this observer, and the final AUC is obtained by averaging the AUCs across observers. There are several reasons for this approach, one being that it is not possible to know whether a difference in ratings between observers is due to differences in interpretation of the rating scale or differences in assessment of the image quality. Thus, determining inter-observer agreement based on the given ratings is not meaningful.

A VGC curve for each class of criteria represents the difference in the assessed image quality between the reference protocol (vender-recommended) and the test protocol (with dose reduction). The area under the VGC curve ($AUC_{VGC}$) is used as a scalar measure of this difference [44]. An $AUC_{VGC}$ not significantly differing from 0.5 for a given criterion indicates similar image quality for that criterion for the two protocols. An $AUC_{VGC}$ significantly larger than 0.5 indicates a higher image quality for the test protocol whereas an $AUG_{VGC}$ significantly smaller than 0.5 indicates a higher image quality for the reference protocol [44,59–61]. The statistical analysis of the $AUC_{VGC}$ was based on the binormal VGC curve, and a fixed-reader analysis was used. Given that the same patients were involved in all comparisons, a paired data analysis was used and a p-value < 0.05 was considered statistically significant. No correction for multiple testing was applied.

## Results

The VGC analysis showed no statistically significant differences in AUCVGC between the protocol with 30% lower dose and the reference (vendor-recommended) protocol for the classes of criteria – demarcation, disturbance, bone structure or tumor homogeneity (Table 3).

**Table 3. The result of the comparison between vendor recommended protocol and the protocol with 30% and 50% dose reduction respectively by using VGC Analyzer.**

| DTS low-dose acquisitions | Vessel demarcation | | | Vessel disturbance | | | Bone structure | | | Tumor homogeneity | | |
|---|---|---|---|---|---|---|---|---|---|---|---|---|
| | $AUC_{VGC}$ | *p*-value | Effect size | $AUC_{VGC}$ | *p*-value | Effect size | $AUC_{VGC}$ | *p*-value | Effect size | $AUC_{VGC}$ | *p*-value | effect size |
| Tomosynthesis acquisition 2 with 30% dose reduction | 0.51 | 0.57 | 0.01 | 0.51 | 0.59 | 0.01 | 0.50 | 0.88 | 0 | 0.47 | 0.09 | 0.03 |
| Tomosynthesis acquisition 3 with 50% dose reduction | 0.47 | 0.03 | 0.03 | 0.47 | 0.02 | 0.03 | 0.50 | 0.79 | 0 | 0.46 | 0.10 | 0.04 |

The VGC analysis demonstrated a small but statistically significant difference in image quality for the classes of criteria demarcation and disturbance, where the protocol with 50% dose reduction was rated lower than the vendor-recommended protocol. For the class of bone structure and tumor homogeneity no significant differences were found between the protocols (Table 3).

Table 3 shows that tumor-homogeneity scores ($AUC_{VGC}$ values) remain close to 0.50 across all participants (all BMI groups) and acquisition protocols, and no statistically significant differences were found between the standard dose protocol and protocols with reduced dose. This means that within this study population, variations in radiation dose and BMI did not materially affect the perceived uniformity or structural assessment of tumors on digital tomosynthesis images. Stable tumor-homogeneity scores indicate that poor image quality due to dose reduction is minimal for the specific task of visualizing tumor structure and texture. As homogeneity is crucial for recognizing abnormal tissue patterns in the background of the lung, consistent scores across protocols imply that diagnostic performance for basic tumor detection is likely maintained, even at lower doses. However, subtle differences in very low contrast or heterogeneous neoplasms may still require confirmation by higher-resolution modalities such as CT.

The effect size for each comparison was quantified by calculating the absolute difference in AUC between each dose-reduced protocol and the standard dose protocol which is 0.5. This effect size demonstrates not only whether differences were statistically significant but also their magnitude, clarifying that observed changes in image quality are statistically detectable but relatively small overall.

## Discussion

This study has compared image quality with pre-defined image quality criteria, between three different DTS protocols, the vendor-recommended protocol and two protocols with radiation dose reduced by 30% and 50%, respectively. In addition, this study aimed to determine whether the DTS technique could be adopted to be included in a lung cancer screening program as a complementary imaging technique.

Our study supports that a dose reduction of 30%, resulting in a dose to the patient of 0.09 mSv per patient, does not lower image quality. A further reduction to 50%, with a dose to the patient of 0.06 mSv, slightly affects image quality, which might be acceptable in a screening context. The dose in CT screening is usually 1–1.5 mSv, thus 10–15 times higher than a screening with tomosynthesis would be. As a large number of healthy individuals are irradiated for every case of cancer that is found, the dose is of importance in screening programs.

This study primarily concerns image quality of thoracic imaging, rather than detection capabilities, which might be considered a limitation. However, DTS could be valuable for follow-up, where image quality is likely the most critical factor.

Adding DTS examinations as follow-up, with shorter intervals, e.g., every 6 months, to a screening program increases population dose – but reducing the dose of the individual examination would minimize this effect. Indeed, if all follow-up CT examinations were replaced by DTS, the total dose from following a lung screening program would be lower.

LDCT is by far the most common method for detecting pulmonary malignancies in screening programs. However, even LDCT has a higher radiation dose, is more expensive, and is less available than DTS. Therefore, radiation dose is of particular importance if DTS would be used in screening as that might have a large impact on population dose and be more cost effective than CT. Reducing the current DTS dose, for example by 30% would bring the dose down close to the dose level in CXR, but DTS provides more information, as a 3D image, than CXR. In addition, the speed of the DTS and the low cost of the equipment means a very low cost per examination.

A drawback of CT lung cancer screening is that it is usually performed every two years, which probably explains why CT screening lowers mortality in lung cancer by only about 20% [9,19,21,22]. In addition, the high costs of previous lung cancer screening programs have so far prevented larger national programs. Therefore, the basic idea is to use DTS more frequently, instead of waiting for the follow-up examination by CT after a two-year interval.

A screening program consisting of an initial LDCT, as it might have higher detection capability as a reference, followed by frequent DTS examinations, would result in a lower total cost per participant. This model could prevent at least as many deaths in lung cancer as previous programs and could be done at lower cost.

It is assumed that a good representation of anatomical structures with DTS images would also result in a good assessment of lung pathology. Similar studies have been explored for assessing image quality of anatomical structures by DTS, but the majority of studies are only aimed at detection of lung cancer or nodules. The studies were designed either as a human observer study by using simulated nodules; an anthropomorphic chest phantom and synthetic nodules, or as a clinical study on patients with nodules graded as clearly visible on DTS images [3,7,40,62–65].

Unlike previous DTS dose-reduction and nodule-detection studies that focused primarily on discrete nodules or limited regions, the current study is the first to prospectively compare three DTS protocols (30% and 50% dose reduction vs the vendor-recommended protocol) in patients with pulmonary malignancies, using multi-reader VGC analysis across thoracic regions. In contrast to earlier studies focused mainly on nodule detectability [4,7,33,37,66–69], our observers graded lung parenchyma and vessels over a wide anatomical extent, including basal, retrocardiac, ventral and peripheral regions, as well as bone structures and tumor homogeneity. This design captures dose effects on overall thoracic image quality rather than on isolated nodules alone, which is crucial if DTS is to be used for interval follow-up and in lung cancer screening programs.

In addition, previous DTS studies, including that of Söderman [40], have mainly assessed dose reduction using simulated noise or phantoms [41,70–72], whereas the current patient study focuses on the systematic visual grading of thoracic image quality at different dose levels. Furthermore, this study systematically investigated pulmonary nodules of all types, not only round nodules, as in many earlier studies, but also irregular lesions with a wide range of shapes and considerable variation in size.

Asplund et al. [41] showed that pulmonary nodule detectability remained unchanged when the effective dose was reduced to approximately 32% of the default setting using noise-simulated tomosynthesis images, which aligns with our finding that diagnostic performance can be maintained despite substantial dose reduction in thoracic DTS. Meltzer et al. [63] later demonstrated that DTS can be used for surveillance of small pulmonary nodules, although measurement variability limits precise growth assessment. In contrast, our prospective patient study directly compares three DTS dose protocols (30% and 50% dose reduction vs the vendor-recommended protocol) and shows that a 30% dose reduction does not degrade observer-rated image quality for vessel demarcation, vessel disturbance, bone structures or tumor homogeneity, while a 50% reduction only slightly worsens vessel-related criteria. These findings extend the existing literature by indicating that routine DTS dose can be lowered towards conventional chest radiography levels without sacrificing clinically relevant image quality, which is particularly important if DTS is to be implemented as a low-dose, low-cost follow-up modality in lung cancer screening pathways and in settings where CT capacity and resources are limited.

Another aspect of this study with relevance beyond its current setting is that many developed and emerging countries face challenges such as limited government support for public healthcare, high insurance costs, and unequal access to advanced diagnostic technologies like CT. LDCT-based lung cancer screening programs have demonstrated reductions

in mortality, but their costs remain high, and many nations are unable to implement lung cancer screening program due to limited resources.

In many low- and middle-income countries, there is a lack of organized and effective screening and prevention programs for cancer due to inadequate infrastructure and limited capacity for program delivery. The major obstacles are late diagnosis and significant delays across the continuum from access to referral and treatment. Key challenges include shortages of trained personnel, inadequate screening services, poor referral procedures, socioeconomic disparities, cultural factors, low public awareness, and long distances to specialized centers [71,72].

For those countries, governments can establish or expand lung cancer screening tomosynthesis technology. Chest tomosynthesis represents an accessible technology that can be implemented with moderate investments, making population-scale screening programs more achievable. The reduced need for CT technical support, lower operating costs, and simplified logistics open opportunities for countries previously excluded from robust screening due to financial or infrastructure limitations. This feature translates to safer, more frequent monitoring for individuals in screening programs, and greater flexibility in implementation. In addition, governments and health systems can adopt shorter follow-up intervals by low-dose DTS without a steep increase in cumulative population dose.

Besides some limitations in this study, the major challenge was the time taken to contact the patients (total 194) and scheduling the examinations due to the pandemic and replacement of the detector at the GE system. Therefore, the data collection took over 2 years (between March 31, 2021 and June 1, 2023).

To start with the limitations, the method of dose reduction required extensive manual labor for example to change the ratio level and attach a 20 mm Al filter in front of the collimator blades between tomosynthesis series. This will not be accepted in a standard clinical operating procedure. Therefore, it probably requires some development, or else improvement is required in the system by the vendor. In addition, the goal of selecting the participants was to find patients with lung lesions detected by CT, which made the selection of potential patients more challenging. Thus, most of the patients were either treated over a long period of time, or had been operated on and needed to be followed up by CT. However, we kept to the standard of a maximum two-week period of time between the CT and DTS examination without any treatment.

Furthermore, there were some potential patients who did not wish to participate in the study, due, for the most part, to their health condition, anxiety after receiving a diagnosis, or anxiety about the upcoming treatment. Finally, it was challenging to involve more than four observers due to the workload of radiologists, although our ambition was to invite radiologists from other hospitals.

The number of included participants may be considered small, but a power calculation before setting up the study showed that 50 participants would be statistically sufficient to assess general image quality with pre-defined criteria on three different DTS protocols. Taking the complexity of this study into account, the number of participants and number of DTS examinations (150) could be considered relatively large when compared to other studies [3,4].

In conclusion, the current study demonstrated that no image quality loss could be identified due to reducing the radiation dose by 30%. This indicates that low-dose DTS could be a possible option to include in lung cancer screening programs.

Replacing LDCT examinations with low-dose DTS in lung cancer screening programs would reduce the radiation dose received by healthy participants by at least 90%. It would also considerably lower costs, as DTS examinations are much faster to perform, and the X-ray equipment is significantly less expensive, an important factor for countries with limited resources or remote areas.

## Acknowledgments

The authors thank Dr. Åke Moritz, Dr Jimmy Yu and Dr. Magnus Ahlbom for evaluation of images, Nasrin Davandy Sarikhan Baglo, for clinical support of the GE system and Ervin Polimac and Michael Toumie for all technical and administrative support with the PACS system.

### Advances in knowledge

The study indicates that there is potential to reduce the current radiation dose in DTS, which would be of significance in a lung cancer screening program.

## Author contributions

**Conceptualization:** Masoud Jadidi, Magnus Båth, Sven Nyrén.

**Data curation:** Masoud Jadidi.

**Formal analysis:** Masoud Jadidi.

**Investigation:** Angelica Svalkvist.

**Methodology:** Masoud Jadidi, Angelica Svalkvist, Magnus Båth.

**Project administration:** Masoud Jadidi.

**Software:** Angelica Svalkvist.

**Supervision:** Sven Nyrén.

**Validation:** Angelica Svalkvist, Magnus Båth, Sven Nyrén.

**Writing – original draft:** Masoud Jadidi, Sven Nyrén.

**Writing – review & editing:** Masoud Jadidi, Magnus Båth, Sven Nyrén.

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
