## [Decision Letter · Decision Letter 0]

10 Nov 2025

PONE-D-25-50559Can the current radiation dose, in chest tomosynthesis, be reduced with retained image quality? A study in the context of lung cancer screening programsPLOS ONE?

Dear Dr. Jadidi,

Thank you for submitting your manuscript to PLOS ONE. After careful consideration, we feel that it has merit but does not fully meet PLOS ONE’s publication criteria as it currently stands. Therefore, we invite you to submit a revised version of the manuscript that addresses the points raised during the review process.

**ACADEMIC EDITOR:**

We look forward to receiving your revised manuscript.

Kind regards,

Aloysius Gonzaga Mubuuke

Academic Editor

PLOS ONE

Journal Requirements:

3. Please ensure that you refer to Figure 1 in your text as, if accepted, production will need this reference to link the reader to the figure.

**Additional Editor Comments:**

The paper talks about an interesting subject in radiology and imaging. The discussion however needs to be strengthened by synthesizing how findings from this study can be useful to a more global audience beyond the setting where the study was conducted from. In addition, the entire paper needs proof-reading to improve on the language.

Reviewers' comments:

Reviewer's Responses to Questions

**Comments to the Author**

1. Is the manuscript technically sound, and do the data support the conclusions?

Reviewer #1: Yes

Reviewer #2: Yes

2. Has the statistical analysis been performed appropriately and rigorously?

Reviewer #1: Yes

Reviewer #2: Yes

3. Have the authors made all data underlying the findings in their manuscript fully available?

Reviewer #1: Yes

Reviewer #2: Yes

4. Is the manuscript presented in an intelligible fashion and written in standard English?

Reviewer #1: Yes

Reviewer #2: Yes

Reviewer #1: Dear author

I have carefully reviewed your manuscript The study addresses an important and clinically relevant question in radiological imaging—whether chest digital tomosynthesis (DTS) dose levels can be reduced while maintaining diagnostic image quality, particularly in the context of lung cancer screening.

Your data convincingly support that a 30% dose reduction in chest DTS preserves subjective image quality across all assessed anatomical and image-quality classes, whereas a 50% reduction introduces mild yet statistically significant degradation in vessel visibility and disturbance.

With improved statistical reporting, clearer explanation of dose-reduction physics, and compliance with data-sharing requirements(Please check below), the manuscript would make a meaningful contribution to the literature on dose optimization in digital tomosynthesis and its potential integration into lung-cancer screening programs.

Thank you for your contribution to this important area of radiological research. I look forward to seeing a revised version of your manuscript that addresses these points.

Sincerely,

1.Acquisition Physics and Dose-Reduction Mechanism:

The reduced-dose protocols were achieved by adding 20 mm aluminium filtration and lowering the tube voltage to 100 kV, while the vendor-recommended protocol is reported as 125 kV. Please clarify this inconsistency (100 kV vs 120 kV/125 kV) and quantify how the added filtration affected beam quality (e.g., half-value layer, mean photon energy). Including measured or simulated x-ray spectra and HVL data would help validate that image-quality changes are not confounded by spectral shifts rather than dose effects.

2. Dose Estimation and Method Transparency

You report that effective dose was calculated from the dose-area product (DAP) of the scout image, scaled by projection ratios and a field-size factor (0.935).

Please detail how these scaling factors were derived, cite supporting references, and provide per-protocol DAP and effective-dose statistics (mean ± SD or median [IQR]) in a summary table. Consider stratifying by patient size or BMI to demonstrate robustness across different body habitus.

3. Statistical Reporting: Precision, Readers, and Multiplicity:

Provide 95% confidence intervals for AUC_{VGC} values and include effect sizes alongside p-values. Also, Quantify inter-observer agreement (e.g., ICC) to assess consistency between readers. You may consider a multi-reader, multi-case (MRMC) analysis or mixed-effects model to appropriately handle correlated data. Finally, discuss whether corrections for multiple testing across the four quality domains were applied.

4.Link to Clinical Performance and Population:

As your cohort consists of patients with known or suspected malignancy rather than a screening population, please temper the generalization to screening settings. If feasible, include an exploratory analysis of lesion visibility stratified by size (e.g., >10 mm vs <10 mm) using CT as the reference. At minimum, discuss how trends in tumour-homogeneity scores may translate to detection or characterization performance.

5.Observer Study Details and Reproducibility:

Clarify the number of rated items per class, treatment of any missing ratings, randomization order, and the contents of the training set.

Indicate whether reading order was balanced across protocols. And, providing your ViewDEX configuration files and full rating criteria as supplementary materials would enhance reproducibility.

Reviewer #2: The authors have presented a well written and comprehensive manuscript based on their study which looks at the possibility of reducing radiation exposure in patients undergoing chest tomosynthesis for the detection of lung cancer. This is a valuable study with potential significance in the clinical setting.

**Do you want your identity to be public for this peer review?** For information about this choice, including consent withdrawal, please see our Privacy Policy

Reviewer #1: **Yes:** ALIYA MULATI

Reviewer #2: No

---

## [Author Response · Author response to Decision Letter 1]

14 Dec 2025

Dear Sir/Madame,

We sincerely thank you for your valuable comments and suggestions, which have contributed significantly to improving the quality of the manuscript. All comments have been carefully addressed, and the manuscript has been revised accordingly.

Best regards

Masoud

---

## [Decision Letter · Decision Letter 1]

21 Jan 2026

PONE-D-25-50559R1Can the current radiation dose, in chest tomosynthesis, be reduced with retained image quality? A study in the context of lung cancer screening programsPLOS One?

Dear Dr. Jadidi,

Thank you for submitting your manuscript to PLOS ONE. After careful consideration, we feel that it has merit but does not fully meet PLOS ONE’s publication criteria as it currently stands. Therefore, we invite you to submit a revised version of the manuscript that addresses the points raised during the review process.

**ACADEMIC EDITOR:** The reviewer has provided valuable insights to improve the paper. In addition, please address the following:

1. The discussion should show some practical implication of your findings in a broad sense such that even readers beyond your institution can get some benefit from the study.

2. How do the findings from your study contribute to known literature around this area?

We look forward to receiving your revised manuscript.

Kind regards,

Aloysius Gonzaga Mubuuke

Academic Editor

PLOS One

Journal Requirements:

Additional Editor Comments :

The reviewer has provided valuable insights to improve the paper. In addition, please address the following:

1. The discussion should show some practical implication of your findings in a broad sense such that even readers beyond your institution can get some benefit from the study.

2. How do the findings from your study contribute to known literature around this area?

3. The paper needs proof-reading to improve the language

Reviewers' comments:

Reviewer's Responses to Questions

**Comments to the Author**

Reviewer #1: All comments have been addressed

2. Is the manuscript technically sound, and do the data support the conclusions?

Reviewer #1: Yes

3. Has the statistical analysis been performed appropriately and rigorously?

Reviewer #1: Yes

4. Have the authors made all data underlying the findings in their manuscript fully available?

Reviewer #1: Yes

5. Is the manuscript presented in an intelligible fashion and written in standard English?

Reviewer #1: Yes

Reviewer #1: Dear Authors,

The revision satisfactorily addresses my prior major points. In particular, the dose-reduction physics and potential spectral confounding are now explicitly handled via spectrum simulation and protocol harmonization across acquisitions. Dose estimation is more transparent, with an explicit DAP→effective-dose coefficient, method citation, and a BMI-stratified summary table that supports robustness across body habitus. Observer-study reproducibility is improved with training-set description, confirmation of no missing ratings, automated randomization, and sharing of ViewDEX configuration files.

I recommend acceptance, with only minor editorial cleanup.

Best,

**Do you want your identity to be public for this peer review?** For information about this choice, including consent withdrawal, please see our Privacy Policy

Reviewer #1: **Yes:** aliya mulati

---

## [Author Response · Author response to Decision Letter 2]

9 Feb 2026

Please see attached file "Response to Reviewers"

---

## [Editor Report · Decision Letter 2]

11 Feb 2026

Can the current radiation dose, in chest tomosynthesis, be reduced with retained image quality? A study in the context of lung cancer screening programs

PONE-D-25-50559R2

Dear Dr. Jadidi,

We’re pleased to inform you that your manuscript has been judged scientifically suitable for publication and will be formally accepted for publication once it meets all outstanding technical requirements.

Kind regards,

Aloysius Gonzaga Mubuuke

Academic Editor

PLOS One

Additional Editor Comments (optional):

None
---

## [Editor Report · Acceptance letter]

PONE-D-25-50559R2

PLOS One

Dear Dr. Jadidi,

I'm pleased to inform you that your manuscript has been deemed suitable for publication in PLOS One. Congratulations! Your manuscript is now being handed over to our production team.

Kind regards,

on behalf of

Dr. Aloysius Gonzaga Mubuuke

Academic Editor

PLOS One